# Neural Relightable Participating Media Rendering

**Quan Zheng**[1,2]**, Gurprit Singh**[1]**, Hans-Peter Seidel**[1]
[1]Max Planck Institute for Informatics, 66123 Saarbrücken, Germany
[2]Institute of Software, Chinese Academy of Sciences, 100190 Beijing, China
`{qzheng, gsingh, hpseidel}@mpi-inf.mpg.de`

## Abstract

Learning neural radiance fields of a scene has recently allowed realistic novel view synthesis of the scene, but they are limited to synthesize images under the original fixed lighting condition. Therefore, they are not flexible for the eagerly desired tasks like relighting, scene editing and scene composition. To tackle this problem, several recent methods propose to disentangle reflectance and illumination from the radiance field. These methods can cope with solid objects with opaque surfaces but participating media are neglected. Also, they take into account only direct illumination or at most one-bounce indirect illumination, thus suffer from energy loss due to ignoring the high-order indirect illumination. We propose to learn neural representations for participating media with a complete simulation of global illumination. We estimate direct illumination via ray tracing and compute indirect illumination with spherical harmonics. Our approach avoids computing the lengthy indirect bounces and does not suffer from energy loss. Our experiments on multiple scenes show that our approach achieves superior visual quality and numerical performance compared to state-of-the-art methods, and it can generalize to deal with solid objects with opaque surfaces as well.

## 1 Introduction

From natural phenomenons like fog and cloud to ornaments like jade artworks and wax figures, participating media objects are pervasive in both real life and virtual content like movies or games. Inferring the bounding geometry and scattering properties of participating media objects from observed images is a long-standing problem in both computer vision and graphics. Traditional methods addressed the problem by exploiting specialized structured lighting patterns [1, 2, 3] or using discrete representations [4]. These methods, however, require the bounding geometry of participating media objects to be known.

Learning neural radiance fields or neural scene representations [5, 6, 7] has achieved remarkable progress in image synthesis. They are able to optimize the representations with the assistance of a differentiable ray marching process. However, these methods are mostly designed for novel view synthesis and have baked in materials and lighting into the radiance fields or surface color. Therefore, they can hardly support downstream tasks such as relighting and scene editing. Recent work [8, 9] has taken initial steps to disentangle the lighting and materials from radiance. For material, their methods are primarily designed for solid objects with opaque surfaces, thus they assume an underlying surface at each point with a normal and a BRDF. The assumed prior, however, does not apply to non-opaque participating media which has no internal surfaces. For lighting, neural reflectance field [8] simulates direct illumination from a single point light, whereas NeRV [9] handles direct illumination and one-bounce indirect illumination. They generally suffer from the energy loss issue due to ignoring the high-order indirect illumination. However, indirect lighting from multiple scattering plays a significant role in the final appearance [10] of participating media.

35th Conference on Neural Information Processing Systems (NeurIPS 2021).

In this paper, we propose a novel neural representation for learning relightable participating media. Our method takes as input a set of posed images with varying but known lighting conditions and designs neural networks to learn a disentangled representation for the participating media with physical properties, including volume density, scattering albedo and phase function parameter. To synthesize images, we embed a differentiable physically-based ray marching process in the framework. In addition, we propose to simulate global illumination by embedding the single scattering and multiple scattering estimation into the ray marching process, where single scattering is simulated by Monte Carlo ray tracing and the incident radiance from multiple scattering is approximated by spherical harmonics (SH). Without supervising with ground-truth lighting decomposition, our method is able to learn a decomposition of direct lighting and indirect lighting in an unsupervised manner.

Our comprehensive experiments demonstrate that our method achieves better visual quality and higher numerical performance compared to state-of-the-art methods. Meanwhile, our method can generalize to handle solid objects with opaque surfaces. We also demonstrate that our learned neural representations of participating media allow relighting, scene editing and insertion into another virtual environment. To summarize, our approach has the following contributions:

1. We propose a novel method to learn a disentangled neural representation for participating media from posed images and it is able to generalize to solid objects.

2. Our method deals with both single scattering and multiple scattering and enables the unsupervised decomposition of direct illumination and indirect illumination.

3. We demonstrate flexibility of the learned representation of participating media for relighting, scene editing and scene compositions.

## 2 Related Work

**Neural scene representations.** Neural scene representations [11, 7, 6] are important building blocks for the recent progress in synthesizing realistic images. Different from representations of components such as ambient lighting and cameras [12, 13, 14] of scenes, neural scene representation [11] learns an embedding manifold from 2D images and scene representation networks [7] aim to infer the 3D context of scenes from images. Classic explicit 3D representations, such as voxels [15, 16, 6], multiplane images [17, 18] and proxy geometry [19] are exploited to learn neural representations for specific purposes. These explicit representations generally suffer from the intrinsic resolution limitation. To sidestep the limitation, most recent approaches shift towards implicit representations, like signed distance fields [20, 21], volumetric occupancy fields [22, 23, 24, 25], or coordinate-based neural networks [26, 7, 27, 28]. By embedding a differentiable rendering process like ray marching [6, 5] or sphere tracing [7, 29] into these implicit representations, these methods are capable of optimizing the scene representations from observed images and synthesizing novel views after training. While they generally show improved quality compared to interpolation based novel view synthesis methods [30, 31], the learned representations are usually texture colors and radiance, without separating lighting and materials. By contrast, we propose to learn a neural representation with disentangled volume density, scattering properties and lighting, which allow the usages in relighting, editing and scene composition tasks.

**Volume geometry and properties capture.** Acquiring geometry and scattering properties of participating media has long been of the interest to the computer vision and graphics community. Early methods utilize sophisticated scanning and recording devices [32, 2] and specialized lighting patterns [3, 1] to capture volume density. Computational imaging methods [33, 34] frame the inference of scattering properties from images as an inverse problem, but they require that the geometries of objects are known. Based on the differentiable path tracing formulation [35], the inverse transport method [36] incorporates a differentiable light transport [37] module within an analysis-by-synthesis pipeline to infer scattering properties, but it aims for known geometries and homogeneous participating media. In contrast, our method learns the geometries and scattering properties of participating media simultaneously and our method can deal with both homogeneous and heterogeneous participating media.

**Relighting.** Neural Radiance Field (NeRF) [5] and its later extensions [38] encode the geometry and radiance into MLPs and leverage ray marching to synthesize new views. While they achieve

realistic results, they are limited to synthesize views under the same lighting conditions as in training. To mitigate this, appearance latent code [39, 40] are used to condition on the view synthesis. Recent approaches [8, 9] decompose materials and lighting by assuming an opaque surface at each point, but this does not apply to participating media. After training, density and materials can be extracted [41] to render new views using Monte Carlo methods [42, 43, 44]. Instead, our method models participating media as a field of particles that scatter and absorb light, which are in accordance with its nature. Neural reflectance field [45, 8] requires collocated cameras and lights during training and simulates only direct illumination. NeRV [9] and OSF [46] simulate merely one-bounce indirect light transport because of the prohibitive computation cost for long paths. However, ignoring the high-order indirect illumination leads to the potential problem of energy loss. By contrast, we use Monte Carlo ray tracing to compute direct lighting and propose to learn a spherical harmonic field for estimating the complete indirect lighting. The PlenOctree [47] uses spherical harmonics to represent the outgoing radiance field as in NeRF, but it does not allow relighting. With both direct illumination and indirect illumination properly estimated, our method enables a principled disentanglement of volume density, scattering properties, and lighting.

## 3 Background

**Volume rendering.** The radiance carried by a ray after its interaction with participating media can be computed based on the radiative transfer equation [48]

$$L_o\left(\boldsymbol{r}_0, \boldsymbol{r}_d\right) = \int_0^\infty \tau\left(\boldsymbol{r}(t)\right) \sigma\left(\boldsymbol{r}(t)\right) L\left(\boldsymbol{r}(t), -\boldsymbol{r}_d\right) \mathrm{d}t. \tag{1}$$

Here, $\boldsymbol{r}$ is a ray starting from $\boldsymbol{r}_0$ along the direction $\boldsymbol{r}_d$ and $\boldsymbol{r}(t) = \boldsymbol{r}_o + t \cdot \boldsymbol{r}_d$ denotes a point along the ray at the parametric distance[1] $t$. $L_o$ is the received radiance at $\boldsymbol{r}_0$ along $\boldsymbol{r}_d$. $\sigma$ denotes the extinction coefficient that is referred to as volume density. The $\tau\left(\boldsymbol{r}(t)\right)$ is the transmittance between $\boldsymbol{r}(t)$ and $\boldsymbol{r}_0$ and it can be computed by $\exp\left(-\int_0^t \sigma(\boldsymbol{r}(s)) \, \mathrm{d}s\right)$. The $L(\cdot)$ inside the integral stands for the in-scattered radiance towards $\boldsymbol{r}_0$ along $-\boldsymbol{r}_d$. NeRF [5] models the in-scattered radiance $L$ (Eq. 1) as a view dependent color $c$, but it ignores the underlying scattering event and incident illumination. Since the learned radiance field of NeRF bakes in the lighting and materials, it allows merely view synthesis under the original fixed lighting, without the support for relighting.

**Ray marching.** The integral in Equation 1 can be solved with the numerical integration method, *ray marching* [49]. This is generally done by casting rays into the volume and taking point samples along each ray to collect volume density and color values [5, 6]. The predicted color of a ray is computed by $L_o\left(\boldsymbol{r}_0, \boldsymbol{r}_d\right) = \sum_j \tau_j \cdot \alpha_j \cdot L\left(\boldsymbol{r}(t_j), -\boldsymbol{r}_d\right)$, where $\alpha_j = 1 - \exp\left(-\sigma_j \cdot \delta_j\right)$, $\delta_j = \|t_{j+1} - t_j\|_2$ and $\tau_j = \prod_{i=1}^{j-1}(1 - \alpha_i)$.

## 4 Neural Relightable Participating Media

In this work, we aim to learn neural representations of participating media with disentangled volume density and scattering properties. We model the participating media as a field of particles that absorb and scatter light-carrying rays. Below we first describe our disentangled neural representation based on a decomposition with single scattering and multiple scattering. Then, we depict our neural network design, followed by details of a volume rendering process for synthesizing images and the details of training.

### 4.1 Lighting Decomposition

We firstly write the in-scattered radiance $L\left(\boldsymbol{r}(t), -\boldsymbol{r}_d\right)$ in Equation 1 as an integral of light-carrying spherical directions over a $4\pi$ steradian range

$$L\left(\boldsymbol{r}(t), -\boldsymbol{r}_d\right) = \int_{\Omega_{4\pi}} S\left(\boldsymbol{r}(t), -\boldsymbol{r}_d, \omega_i\right) L_{in}(\boldsymbol{r}(t), \omega_i) \, \mathrm{d}\omega_i, \tag{2}$$

---

[1]While the integration accounts for a $t$ going to $\infty$, $t$ covers only the range with participating media in practice.

where $L_{in}$ is the incident radiance at $\boldsymbol{r}(t)$ from the direction $\omega_i$. $S(\cdot)$ is a scattering function which determines the portion of lighting that is deflected towards the $-\boldsymbol{r}_d$. Previous methods [8, 9, 41] assume a surface prior with a normal at every point and account for $2\pi$ hemispherical incident directions. Accordingly, they define the scattering function $S$ as a BRDF function. This assumption, however, can hardly match the participating media objects which have no internal surfaces and normals. By contrast, we deal with light-carrying directions over the full $4\pi$ steradian range. Specifically, we define $S = a(\boldsymbol{r}(t)) \cdot \rho(-\boldsymbol{r}_d, \omega_i, g)$ to account for the scattering over the full spherical directions. Here, $a(\boldsymbol{r}(t))$ is the scattering albedo. $\rho$ is the Henyey-Greenstein (HG) [50] phase function[2] that decides the scattering directionality (Appendix A), where $g$ is an *asymmetry parameter* in $(-1, 1)$. For brevity, we omit $g$ in the notation of $\rho$. Then, we can rewrite the radiance of ray $\boldsymbol{r}$ as the disentangled form:

$$L_o(\boldsymbol{r}_0, \boldsymbol{r}_d) = \int_0^\infty \tau(\boldsymbol{r}(t)) \sigma(\boldsymbol{r}(t)) \int_{\Omega_{4\pi}} a(\boldsymbol{r}(t))\rho(-\boldsymbol{r}_d, \omega_i)L_{in}(\boldsymbol{r}(t), \omega_i)\, \mathrm{d}\omega_i\, \mathrm{d}t, \quad (3)$$

from which the volume density $\sigma$ decides the geometries, and the albedo $a$ along with the phase function $\rho$ controls the scattering of rays. We propose to train neural networks to learn the volume density and scattering properties.

To compute $L_o$, we additionally need to estimate $L_{in}$ (Eq. 3). This can be conducted by recursively substituting the $L_o$ into $L_{in}$ and expressing the radiance for a pixel $k$ as an integral over all paths $P_k = \int_\Psi g_k(\bar{x})d\mu(\bar{x})$, where $\bar{x}$ is a light-carrying path, $g_k$ denotes a measurement contribution function, $\mu$ is the measure for the path space, and $\Psi$ is the space of paths with all possible lengths. $P_k$ is then computed by a summation of the contributions of all paths. The contribution of a path with length $i$ can be written as

$$P_{k,i} = \int_{l_1} \int_\Omega \cdots \int_{l_{i-1}} \int_\Omega L_e(x(t_{i-1})) V(x(t_{i-1}), \omega_{i-1}) \mathcal{T}(P_{k,i})\, \mathrm{d}t_1\, \mathrm{d}\omega_1 \cdots \mathrm{d}t_{i-1}\, \mathrm{d}\omega_{i-1}, \quad (4)$$

where $x(t_{i-1})$ is a point along the $(i-1)$-th ray segment with the length $l_{i-1}$, $\omega_{i-1}$ is the scattering direction in the space $\Omega$ and $\omega_0 = \boldsymbol{r}_d$, $L_e$ denotes the emitted radiance towards $x(t_{i-1})$ from a light source, $V$ accounts for the transmittance between $x(t_{i-1})$ and the light source, $\mathcal{T}(P_{k,i}) = \prod_{j=1}^{i-1} V(x(t_j), \omega_j)\sigma(x(t_j)) \cdot \prod_{j=1}^{i-1} \rho(-\omega_{j-1}, \omega_j)a(x(t_j))$ is called the path throughput. Volumetric path tracing [44] utilizes Monte Carlo method to estimate the integral but its computational cost increases quickly when using high sampling rates of paths with many bounces. To reduce the cost, NeRV [9] truncates the paths and only considers up to one indirect bounce, namely $i = 3$ in Eq. 4. This, however, leads to energy loss since high-order indirect illumination are neglected.

Instead of tracing infinitely long paths or truncating the paths, we propose to decompose the in-scattered radiance $L$ in Eq. 1 as $L = L_s + L_m$, where $L_s$ is the single-scattering contribution and $L_m$ denotes the multiple-scattering contribution. Therefore, $L_o$ can be split into two integrals: $L_{o,s} = \int_0^\infty \tau(\boldsymbol{r}(t)) \sigma(\boldsymbol{r}(t)) L_s\, \mathrm{d}t$ and $L_{o,m} = \int_0^\infty \tau(\boldsymbol{r}(t)) \sigma(\boldsymbol{r}(t)) L_m\, \mathrm{d}t$, that can be evaluated separately.

**Single scattering.** To compute the $L_{o,s}$, we evaluate the following integral at each sample point $\boldsymbol{r}(t)$ along the camera ray $\boldsymbol{r}$

$$L_s = \int_{\Omega_{4\pi}} a(\boldsymbol{r}(t))\rho(-\boldsymbol{r}_d, \omega_i)L_e(\boldsymbol{r}(t), \omega_i)V(\boldsymbol{r}(t), \omega_i)\, \mathrm{d}\omega_i. \quad (5)$$

Here, $L_e$ is the emitted radiance from a light source to the point $\boldsymbol{r}(t)$ and $V$ is the transmittance between $\boldsymbol{r}(t)$ and a light source. The transmittance can be computed with another integral of volume density as described in Section 3, but computing the integral for all points leads to high computation cost during training and inference. Therefore, we train a visibility neural network to regress the transmittance value as done in [9].

**Multiple scattering.** For $L_{o,m}$, we evaluate the $L_m = \int_{\Omega_{4\pi}} a(\boldsymbol{r}(t))\rho(-\boldsymbol{r}_d, \omega_i)L_{in}(\boldsymbol{r}(t), \omega_i)d\omega_i$. $L_m$ aggregates the incoming radiance of rays that have been scattered at least once in the participating media. Since the distribution of incident radiance from multiple scattering is generally smooth, we propose to represent the incident radiance $L_{in}$ as a spherical harmonics expansion: $L_{in}(\omega_i) = $

---

[2]Both the incident and outgoing directions point away from a scattering location in this paper.

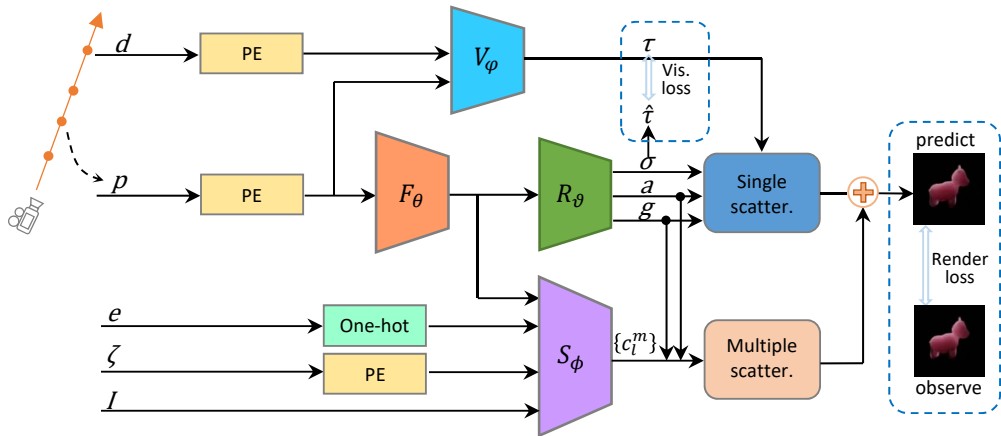

Figure 2: Our overall architecture for learning neural participating media. "PE" denotes the positional encoding and "one-hot" denotes the one-hot encoding. The rendering loss and the visibility loss correspond to the summands in Eq. 9.

$\mathcal{F}\left(\sum_{l=0}^{l_{max}}\sum_{m=-l}^{l} c_l^m Y_l^m(\omega_i)\right)$, where $l_{max}$ is the maximum spherical harmonic *band*, $c_l^m \in \mathbb{R}^3$ are spherical harmonic coefficients for the RGB spectrum, $Y_l^m$ are spherical harmonic basis functions and $\mathcal{F}(x) = \max(0, x)$. Therefore, we compute the multiple-scattering contribution with

$$L_m = \int_{\Omega_{4\pi}} a(\boldsymbol{r}(t))\rho(-\boldsymbol{r}_d, \omega_i) \cdot \mathcal{F}\left(\sum_{l=0}^{l_{max}}\sum_{m=-l}^{l} c_l^m Y_l^m(\omega_i)\right) d\omega_i. \qquad (6)$$

We employ a neural network to learn spherical harmonic coefficients. By using spherical harmonics for the incident radiance from multiple scattering, we sidestep the lengthy extension of the path integral (Eq. 4). Since we introduce the approximation of multiple scattering at the primary rays, we sidestep the explosion of rays. Figure 1 visualizes the explosion of rays when computing indirect illumination under an environment lighting. Multiple shadow rays are needed to account for the directional emission from the light source. The brute-force ray splitting approach leads to explosion of rays and is impractical. NeRV reduces the ray count by tracing up to one indirect bounce, but it has a complexity of $\mathcal{O}(M \cdot N)$, where $M$ is the number of first indirect bounces (red) and $N$ is the number of shadow rays (blue). Our method uses spherical harmonics to handle indirect illumination as a whole and its complexity is $\mathcal{O}(k \cdot M)$, where $k$ is the sampling rates along the primary rays.

**Network architectures.** Figure 2 presents our overall architecture. Our neural networks are based on the coordinate-based MLP strategy, and we use frequency-based positional encoding [5, 51] $E$ to map an input coordinate $p$ to a higher dimensional vector $E(p)$ before sending it to the neural networks. Specifically,

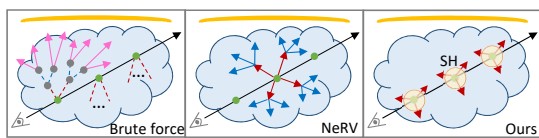

Figure 1: Visualization of path bounces.

we use the MLP $R_\vartheta$ to predict volume density $\sigma$ (1D), scattering albedo $a$ (3D), and *asymmetry parameter* $g$ (1D). Meanwhile, we employ the MLP $S_\phi$ to learn spherical harmonic coefficients $c_l^m$. Here, $S_\phi$ is conditioned on the point light location $\zeta$, the point light intensity $I$, and a binary indicator $e$ which is one-hot encoded to indicate the existence of environment lighting. Since both the property network $R_\vartheta$ and the SH network $S_\phi$ takes as input the encoded coordinate, we introduce a feature network $F_\theta$ to predict shared features for downstream MLPs. To get the visibility for shadow rays, we train another MLP $V_\varphi$, which takes in the encoded direction $d$ in addition to the encoded coordinate $p$, to learn visibility values for the estimation of single scattering. In summary, we have

$$R_\vartheta : F_\theta(E(p)) \rightarrow (\sigma, a, g) \quad S_\phi : (e, \zeta, I, F_\theta(E(p))) \rightarrow \{c_l^m\} \quad V_\varphi : (\boldsymbol{r}(t), E(d)) \rightarrow \tau. \qquad (7)$$

Note that the above $R_\vartheta$ learns per-location *asymmetry parameter* $g$ (Appendix A). Yet, for scenes with a single participating media object, we use a singe $g$ and optimize it during training.

## 4.2 Volume Rendering

Based on the above decomposition, we employ *ray marching* (Sec. 3) to numerically estimate $L_{o,s}$ and $L_{o,m}$. Hence, the final radiance $L_o$ of the camera ray $r$ in Eq. 1 can be computed as:

$$L_o(r) = \Sigma_{j=1}^{N} \tau(r(t)) (1 - \exp(-\sigma(r(t)) \cdot \delta t_j)) (L_s + L_m), \qquad (8)$$

where we sample $N = 64$ query points in a stratified way along the ray $r$ and $\delta t_j = \|r(t_{j+1}) - r(t_j)\|_2$ is the step size. For each point sample, we query the MLPs to obtain its scattering properties and SH coefficients for computing single scattering and multiple scattering.

We compute single scattering at a point based on Eq. 5. We shoot shadow rays towards light sources to get the emitted radiance $L_e$ according to the light types. For environment lighting, we sample 64 directions stratified over a sphere around the point to obtain incident radiance. For a point light, we directly connect the query point to the light source. To account for the attenuation of light radiance, we query $V_\varphi$ to get the visibility to the light source from the query point.

For multiple scattering, we uniformly sample $K = 64$ random incident directions over the sphere around each query point, evaluate the incident radiance $L_{in}$ along each direction using the learned spherical harmonic coefficients, and estimate the integral in Eq. 6 with a Monte Carlo integration $L_m = 1/K \sum_{i=1}^{K} a\rho(\omega_i)L_{in}(\omega_i)$. For brevity, we omit the $r$ notation. Note that the visibility towards the light source is not needed in the computation.

## 4.3 End-to-end Learning

Based on the fully differentiable pipeline, we can end-to-end learn a neural representation for each scene. The learning requires a set of posed RGB images and their lighting conditions. During each training iteration, we trace primary rays through the media. Along each primary ray, we estimate single scattering using shadow rays and compute multiple scattering contribution via the learned spherical harmonic coefficients as described in Sec. 4.2. We optimize the parameters of $F_\theta$, $R_\vartheta$, and $S_\phi$ by minimizing a rendering loss between the predicted radiance $L_o(r)$ from *ray marching* and the radiance $\hat{L}_o(r)$ from input images. To train the visibility network $V_\varphi$, we use the transmittance $\hat{V}_\vartheta$ computed from the learned volume density as the ground truth and minimize the visibility loss between the prediction $V_\varphi$ and the ground truth. Therefore, our loss function includes two parts:

$$\mathcal{L} = \sum_{r \in \mathcal{R}} \|\Gamma(L_o(r)) - \Gamma(\hat{L}_o(r))\|_2^2 + \mu \cdot \sum_{r \in \mathcal{R}, t} \|V_\varphi(r(t), r_d) - \hat{V}_\vartheta(r(t), r_d)\|_2^2, \qquad (9)$$

where $\Gamma(L) = L/(1+L)$ is a tone mapping function, $\mathcal{R}$ is a batch of camera rays and $\mu = 0.1$ is the hyperparameter to weight the visibility loss.

## 4.4 Implementation Details

Our feature MLP $F_\theta$ has 8 fully-connected ReLU (FC-ReLU) layers with 256 channels per layer. The downstream $S_\phi$ consists of 8 FC-ReLU layers with 128 channels per layer, whereas the $R_\vartheta$ uses one such layer with 128 channels. The visibility MLP $V_\varphi$ has 4 FC-ReLU layers with 256 channels per layer. We set the maximum positional encoding frequency to $2^8$ for coordinates $p$, $2^1$ for directions $d$, and $2^2$ for the 3D location of the point light.

We train all neural networks together to minimize the loss (Eq. 9). We use the Adam [52] optimizer with its default hyperparameters and schedule the learning rate to decay from $1 \times 10^{-4}$ to $1 \times 10^{-5}$ over 200K iterations. For each iteration, we trace a batch of 1200 primary rays. Note we stop the gradients from the visibility loss to the property network and the feature network so that they do not compromise the learning to match the visibility network.

# 5 Experiments

We firstly evaluate our method by comparing it with state-of-the-art methods on simultaneous relighting and view synthesis. Then, we demonstrate that our learned neural representations allow flexible editing and scene compositions, followed by ablation studies of this approach. Please refer to the appendices for additional results.

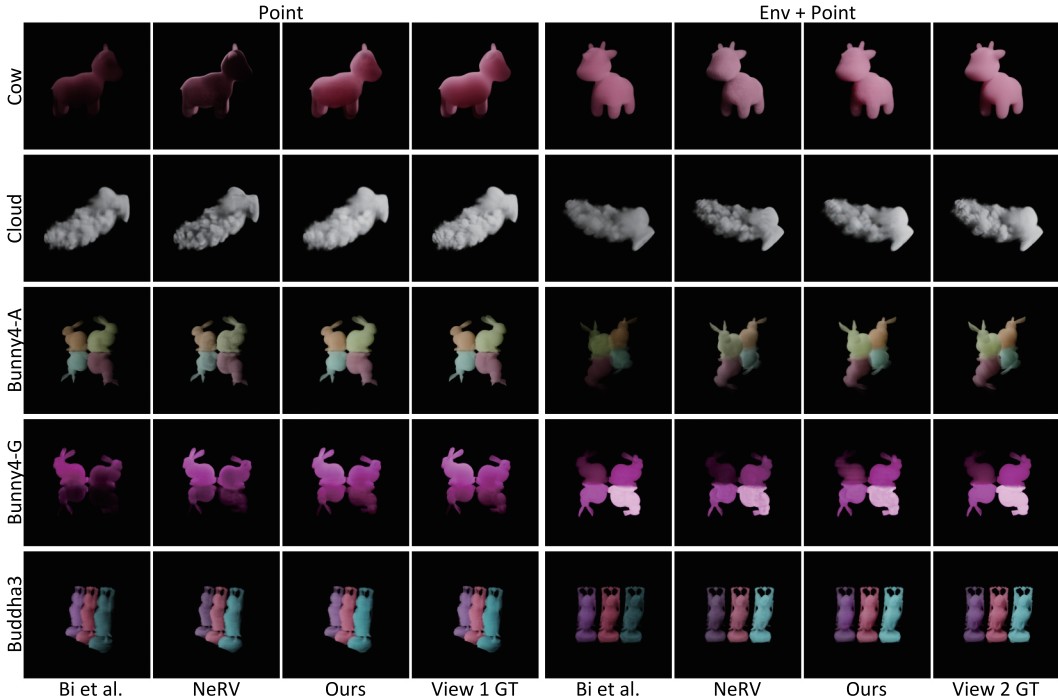

Figure 3: Qualitative comparisons of simultaneous view synthesis and relighting. The training illumination for the left half is "point" which contains a single point light. The training illumination for the right half is "env + point". GT denotes the ground truth image.

## 5.1 Experiment Settings

**Compared methods.** We compare our method with state-of-the-art baselines [8, 9]. They are designed for scenes with solid objects and do not trivially extend to handle participating media, so we implement them with the new functionality to handle participating media. Please refer to the Appendix D for the implementation details.

**Datasets.** We produce datasets from seven synthetic participating media scenes. The *Cloud* scene is heterogeneous media and the others are homogeneous media. *Bunny4-VaryA* and *Buddha3* are set with spatially varying albedo. *Bunny4-VaryG* and *Buddha3* are configured with spatially varying asymmetry parameters. Each scene is individually illuminated with two lighting conditions. The first one "point" has a white point light with varying intensities sampled within $50 \sim 900$ and its location is randomly sampled on spheres with the radius ranging from 3.0 to 5.0; The second one "env + point" contains a fixed environment lighting and a randomly sampled white point light. Each dataset contains 180 images, from which we use 170 images for training and the remaining for validation. In addition, we prepare a test set with 30 images for each scene to test the trained models. Each test image is rendered with a new camera pose and a new white point light that is located on a sphere of the radius $4.0$. Since Bi's method [8] requires a collocated camera and light during training, we additionally generated such datasets for it. For the "env + point" datasets, we randomize the usage of environment lighting across images and record a binary indicator for each image.

## 5.2 Results

**Relighting comparisons.** We show the qualitative comparisons of simultaneous relighting and view synthesis on the test data in Fig. 3. The left half is trained with the "point" lighting condition, whereas the right half is trained with the "env + point". Bi's method shows artifacts in each case as it has no mechanisms to simulate the multiple scattering that significantly affects the appearance of participating media. NeRV handles the environment illumination properly but shows artifacts on the participating media objects. Our method achieves realistic results on all test sets with

Table 1: Quantitative comparisons on the test data for training on the "point" illumination. We measure image qualities with PSNR (↑), SSIM (↑) and ELPIPS (↓) [53]. ELPIPS values below have a scale of $\times 10^{-2}$. Note the tabulated values are the mean values over all images of a test set.

| Point | Cow | | | Cloud | | | Bunny4-VaryA | | | Bunny4-VaryG | | | Buddha3 | | |
|---|---|---|---|---|---|---|---|---|---|---|---|---|---|---|---|
| Method | PSNR | SSIM | ELPIPS | PSNR | SSIM | ELPIPS | PSNR | SSIM | ELPIPS | PSNR | SSIM | ELPIPS | PSNR | SSIM | ELPIPS |
| Bi et al. | 24.70 | 0.958 | 0.465 | 20.92 | 0.921 | 0.783 | 27.29 | 0.960 | 0.378 | 29.40 | 0.971 | 0.334 | 29.47 | 0.970 | 0.299 |
| NeRV | 25.20 | 0.960 | 0.540 | 25.68 | 0.949 | 0.526 | 27.67 | 0.969 | 0.306 | 26.76 | 0.968 | 0.419 | 28.69 | 0.969 | 0.315 |
| Ours | **34.20** | **0.983** | **0.184** | **33.51** | **0.974** | **0.302** | **34.75** | **0.980** | **0.189** | **33.86** | **0.981** | **0.257** | **33.77** | **0.975** | **0.245** |

Table 2: Quantitative comparisons on the test data for training on the "env + point" datasets.

| Env+Point | Cow | | | Cloud | | | Bunny4-VaryA | | | Bunny4-VaryG | | | Buddha3 | | |
|---|---|---|---|---|---|---|---|---|---|---|---|---|---|---|---|
| Method | PSNR | SSIM | ELPIPS | PSNR | SSIM | ELPIPS | PSNR | SSIM | ELPIPS | PSNR | SSIM | ELPIPS | PSNR | SSIM | ELPIPS |
| Bi et al. | 24.84 | 0.960 | 0.501 | 22.18 | 0.934 | 0.709 | 26.65 | 0.958 | 0.464 | 30.03 | 0.974 | 0.285 | 23.41 | 0.938 | 0.679 |
| NeRV | 27.83 | 0.974 | 0.413 | 26.07 | 0.950 | 0.476 | 28.18 | 0.968 | 0.301 | 27.97 | 0.975 | 0.339 | 28.99 | 0.969 | 0.299 |
| Ours | **33.32** | **0.982** | **0.209** | **32.64** | **0.969** | **0.353** | **34.47** | **0.979** | **0.191** | **34.09** | **0.982** | **0.243** | **34.03** | **0.975** | **0.261** |

either homogeneous media or heterogeneous media. Table 1 and Table 2 present the corresponding quantitative measurements, where our method overtakes the compared methods on each test set.

Using the same batch size, our training with 200K iterations on a Nvidia Quadro RTX 8000 GPU takes one day, whereas Bi's method and NeRV takes 22h and 46h. For a $400 \times 400$ image, our average inference time is 7.9s, while Bi's method and NeRV takes 53.2s and 21.9s, respectively.

**Learned lighting decomposition.** Without using any ground-truth lighting decomposition data, our method is able to learn the decomposition of lighting in an unsupervised way.

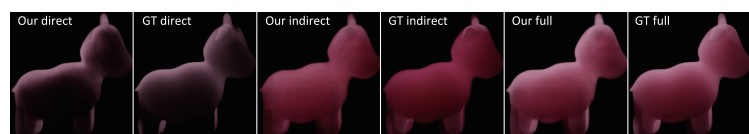

Figure 4: Lighting decompositions.

Figure 4 presents our decomposed results on a test view of the *Cow* scene, with the single-scattering component (direct lighting) and the multiple-scattering component (indirect lighting), and the corresponding ground-truth images.

**Scene editing and scene composition.** Our method learns neural representations for the participating media scenes. After training, we can query the neural networks to obtain the volume density, the albedo, and the phase function parameter. This allows flexible editing to achieve desired effects or insertion into a new virtual environment for content creation. In addition, we can leverage a standard rendering engine to render these data. Figure 5 compares the rendering

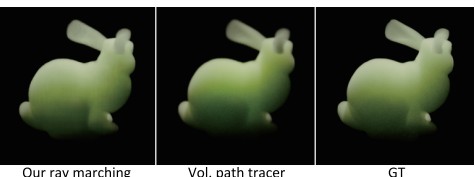

Figure 5: Ray marching vs. volume path tracing.

of the learned *Bunny* with ray marching using the neural network and with a volumetric path tracer. To render with the path tracer, we first queried the neural network to obtain 128 x 128 x 128 data volumes of volume density and albedo. Both rendered results are visually similar to the ground truth. Figure 6 demonstrates an editing of the red channel of the albedo to achieve the red cloud and another editing of the volume density to make the cloud thinner.

We show in Fig. 7 that we can compose a scene consisting of our learned cow and a gold sculpture described by traditional meshes and materials (Fig. 7 bottom). Similarly, we can construct a scene composed entirely of our learned objects (Fig. 7 top). To render the composed scenes, we slice out discrete volumes with a resolution $128 \times 128 \times 128$ from the volume density field and the albedo field and conduct the Monte Carlo rendering in Mitsuba [54].

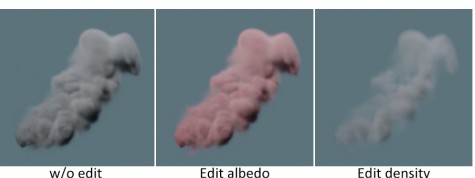

Figure 6: Edit the learned cloud.

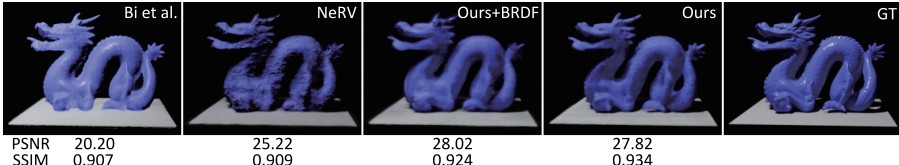

| | | | |
|---|---|---|---|
| Bi et al. | NeRV | Ours+BRDF | Ours | GT |

| | | | | |
|---|---|---|---|---|
| PSNR | 20.20 | 25.22 | 28.02 | 27.82 |
| SSIM | 0.907 | 0.909 | 0.924 | 0.934 |

Figure 8: Comparisons on a test view of the solid *Dragon* with glossy surfaces. PSNR and SSIM metrics for this view are listed below images. The training illumination is from a single point light and the test illumination is a new white point light.

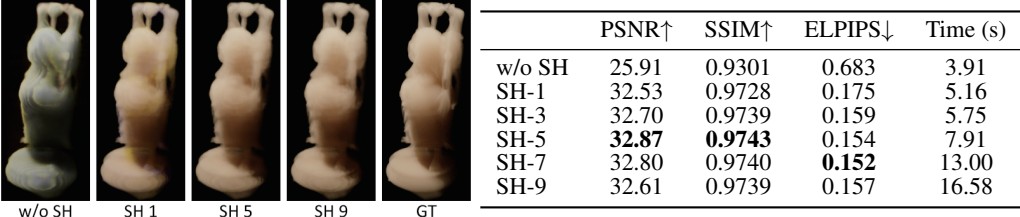

| | PSNR↑ | SSIM↑ | ELPIPS↓ | Time (s) |
|---|---|---|---|---|
| w/o SH | 25.91 | 0.9301 | 0.683 | 3.91 |
| SH-1 | 32.53 | 0.9728 | 0.175 | 5.16 |
| SH-3 | 32.70 | 0.9739 | 0.159 | 5.75 |
| SH-5 | **32.87** | **0.9743** | 0.154 | 7.91 |
| SH-7 | 32.80 | 0.9740 | **0.152** | 13.00 |
| SH-9 | 32.61 | 0.9739 | 0.157 | 16.58 |

w/o SH    SH 1    SH 5    SH 9    GT

Figure 9: Image quality and mean inference timings of different number of spherical harmonic bands. ELPIPS metrics have a scale of $10^{-2}$. The full images, including the SH-3 and SH-7, are documented in Appendix E.

**Scene of solid objects.** Beyond the scenes with participating media, our method can be used for scenes with solid objects. Figure 8 shows a comparison between our method and the baselines on the *Dragon* scene which contains glossy opaque surfaces. "Ours+BRDF" is a variant of our method that uses SH for indirect illumination, but adopts a classical BRDF model [55] and trains the neural network to predict parameters of the BRDF model as in [8, 9]. Bi's method produces an overexposed appearance and the shadow on the floor gets faint. Our method achieves a smooth appearance and higher numerical metrics, whereas "Ours + BRDF" recovers the highlights on the glossy dragon better.

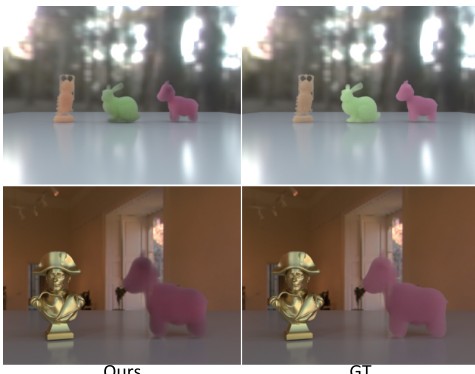

Ours                GT

Figure 7: Scene compositions.

### 5.3 Ablation Studies

**Spherical harmonic bands.** We analyze the effect of the maximum spherical harmonics band $l_{max}$ of Eq. 6 based on the *Buddha* scene. Figure 9 compares the same test view of each case on the left and tabulates the average quality measurements over the test set on the right. Removing the spherical harmonics from our method leads to quality drop and color shift is observed in its result. Based on the numerical metrics and inference timings, we select the $l_{max} = 5$ for other experiments.

**Scattering function.** We show in Fig. 8 that our method with the HG phase function can be applied to a scene of solid objects. In addition, we conduct an ablation by applying the variant "Ours+BRDF" to the *Bunny* scene of participating media. Figure 10 shows that the variant has difficulty in learning the volume density and leads to many cracks, while the proposed method performs robustly on this scene.

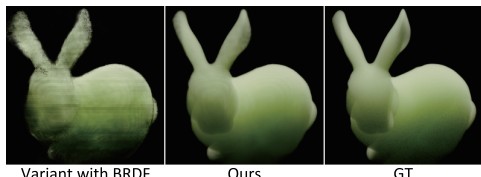

Variant with BRDF        Ours        GT

Figure 10: Compare Ours+BRDF and Ours with phase function.

## 6 Limitations and Future Work

**Real-world scenes.** In this work, we learn neural representations from synthetic datasets with varying but known lighting conditions. Also, the camera

poses are available to the method. It would be interesting to extend this method to handle participating media captured from real-world scenes with unknown lighting conditions and unknown camera poses. In that case, the illumination and camera poses of the scenes need to be estimated in the first place.

**Glossy reflections.** For scenes with glossy solid objects (Fig. 8), our method tends to reproduce a smooth appearance and the glossy highlights are not as sharp as the ground truth. An avenue for future research would be to develop methods to recover the glossy reflections better.

**Generalizability.** Our ray marching with the trained neural network generalizes well to unseen light intensities and light locations that are in the range of the training data. That said, its generalization is in an interpolation manner. For light intensities and light locations that are outside of the training range, the generalization quality of the neural network gradually decreases. Please refer to the Appendix H for the analysis on the generalization quality.

**Media within refractive boundaries.** Our method achieves realistic results for participating media without refractive boundaries, like cloud, fog, and wax figure. Applying our method to participating media within refractive boundaries, like wine in a glass, entails further work as the refractive boundaries cause ambiguities due to deflecting the camera rays and the light rays.

## 7    Conclusion

We have proposed a novel method for participating media reconstruction from observed images with varying but known illumination. We propose to simulate direct illumination with Monte Carlo ray tracing and approximate indirect illumination with learned spherical harmonics. This enables our approach to learn to decompose the illumination as direct and indirect components in an unsupervised manner. Our method learns a disentangled neural representation with volume density, scattering albedo and phase function parameters for participating media, and we demonstrated its flexible applications in relighting, scene editing and scene compositions.

## Acknowledgments and Disclosure of Funding

We acknowledge the valuable feedback from reviewers. This work was supported by Research Executive Agency 739578 and CYENS Phase 2 AE739578.

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
