# Appendices

## A   Phase Function Details

Our phase function for participating media (Sec. 4.1) is the Henyey-Greenstein (HG) function [1]

$$p(\omega_o, \omega_i, g) = \frac{1}{4\pi} \frac{1 - g^2}{(1 + g^2 + 2g\cos\theta)^{3/2}}, \tag{1}$$

where $\theta$ is the angle between the outgoing direction $\omega_o$ and the incident direction $\omega_i$. For the notations of directions, we use the convention that both the incident and outgoing rays point away from a scattering location. $g$ is called the *asymmetry parameter* that is in $(-1, 1)$. A positive $g$ value is for forward scattering, a negative $g$ value is for backward scattering, and the zero value is for isotropic scattering.

Same as in [2, 3], our BRDF function for the experiment on solid objects (Sec. 5.2) is the analytical model [4] which combines a specular component using the ggx distribution [5] and a diffuse component.

## B   Spherical Harmonics Details

In Sec. 4.1 of the main paper, we propose to represent the incident radiance due to multiple scattering with spherical harmonics. Spherical Harmonics (SH) are orthonormal basis defined on complex numbers over the unit sphere. Since our radiance function is defined in the real number domain, our SH basis functions $Y_l^m(\omega_i)$ $(0 \le l \le l_{max}, -l \le m \le l)$ in Eq. 6 (main paper) are *real* spherical harmonic functions

$$Y_l^m(\theta_i, \phi_i) = \begin{cases} \sqrt{2} K_l^m \cos(m\phi_i) P_l^m(\cos\theta_i) & m > 0 \\ K_l^m P_l^m(\cos\theta_i) & m = 0 \\ \sqrt{2} K_l^m \sin(-m\phi_i) P_l^{-m}(\cos\theta_i) & m < 0, \end{cases} \tag{2}$$

where $(\theta_i, \phi_i)$ are the spherical coordinates of the direction $\omega_i$ that is in the Cartesian coordinate system, $K_l^m = \sqrt{\frac{(1+2l)}{4\pi} \frac{(l-|m|)!}{(l+|m|)!}}$ is a normalization factor, and $P_l^m$ are the associated Legendre polynomials. Then, the incident radiance function $\widetilde{L}(\omega_i)$ can be computed using the SH basis

$$\widetilde{L}(\omega_i) = \mathcal{F}\left(\sum_{l=0}^{l_{max}} \sum_{m=-l}^{l} c_l^m Y_l^m(\omega_i)\right), \tag{3}$$

where $c_l^m \in \mathbb{R}^3$ are SH coefficients, $l_{max}$ is the maximum SH band, and $\mathcal{F}(x) = \max(0, x)$ ensures non-negative incident radiance.

In each training iteration, we sample $K = 64$ random incident directions $\{\omega_i\}_{i=1}^{K}$ and evaluate the $Y_l^m(\omega_i)$. To reduce the computational cost, we reuse these incident directions and the evaluated $Y_l^m(\omega_i)$ for all point samples along the primary rays of this batch.

## C   Additional Implementation Details of Our Method

**Training details.**   We end-to-end train our model to learn a separate neural representation of each scene. In each training iteration, we randomly draw a batch of 1200 primary rays across all training views. Our visibility network is trained to match the learned scene geometry by the property network, so it is optimized according to the visibility values computed from the volume density, without requiring ground-truth visibility. We cut off the gradient from the *render loss* to the visibility network. Meanwhile, we cut off the gradient from the *visibility loss* to the property network so that it does not degrade its learning of the volume density.

**Inference details.**   Our inference uses the same setting as the training. We draw 64 point samples along each camera ray to query our model. The number of incident directions for computing indirect illumination is 64. We set the number of shadow rays to 1 for the point light, whereas we use 32 shadow rays for the environment lighting.

## D   Implementation details of the two baselines

For our comparisons, we implemented the Neural Reflectance Field [2] and NeRV [3] as our baselines. Since they were designed for scenes with solid objects, we adapt them to cope with participating media.

Our implementation of the Neural Reflectance Field [2] baseline uses the same neural network architecture and positional encoding as in the original paper. Specifically, we implement the dual-network design with a coarse network and a fine network. Each neural network is an MLP consisting of 14 fully-connected ReLU layers with 256 neurons per layer. Also, we apply frequency-based positional encoding to transform the input 3D coordinates with a maximum frequency $2^{10}$. Different from the eight-channel output in the original paper, we have a four-channel output consisting of a 1-D volume density and a 3-D scattering albedo. Along each ray, we draw 64 stratifed point samples for the coarse network and 128 point samples for the fine network.

In our implementation of the NeRV [3] baseline, we utilize an MLP with 8 fully-connected ReLU layers to compute the physical properties. Each layer has 256 neurons. In addition, we employ a visibility MLP [3] to compute a 1-D visibility and a 1-D expected termination depth. The visibiliy MLP firstly processes the encoded coordinates using 8 fully-connected ReLU layers with 256 neurons per layer to get an 8-D output. The output, concatenated with the encoded directions, is further processed by 4 fully-connected ReLU layers with 128 neurons per layer. The maximum positional encoding frequencies for 3D coordinates and directions are $2^7$ and $2^4$, respectively. Along each camera ray, we take 64 stratified samples, same as in our method. We trace one shadow ray for the point light source and 32 shadow rays for the environment lighting. For the "point" illumination, we uniformly take 128 random directions for the first indirect bounces at the termination depth along a ray. For the "env + point" illumination, we set the number of first indirect bounces to 32.

## E   Additional images for the spherical harmonic band ablation

In the ablation study of the maximum spherical harmonic band (Sec. 5.3), we show the results of SH-1, SH-5, and SH-9. We present the full comparison with SH-3 and SH-5 in Fig. 1. The quantitative metrics are listed in the right table of Fig. 9 (Sec. 5.3 in the main paper). Note that SH-5 achieves the qualitative result that is similar to SH-7 and SH-9, but SH-5 gives higher numerical performance than others (Fig. 1).

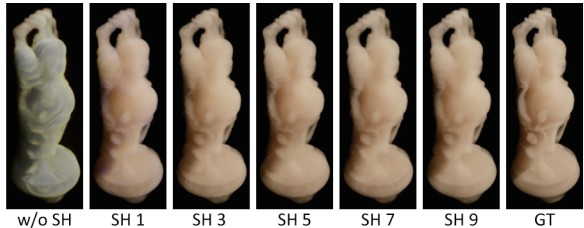

Figure 1: Qualitative comparisons of different settings of the maximum spherical harmonic band from SH-1 to SH-9. "w/o SH" denotes without using spherical harmonics and "GT" denotes the ground truth.

## F   Additional quantitative results for participating media scenes

### F.1   Scenes trained on the "point"

Table 1 presents additional numerical results for the single *Bunny* and *Buddha* scenes when using the "point" training illumination. Each test view has a new point light. The tabulated values are the mean values over images of the test set. Our approach outperforms the two baselines quantitatively.

Table 1: Quantitative comparisons on the test data. Image qualities are measured with PSNR ($\uparrow$), SSIM ($\uparrow$) and ELPIPS ($\downarrow$) [6]. ELPIPS values have a scale of $\times 10^{-2}$.

| Point | Bunny | | | Buddha | | |
|---|---|---|---|---|---|---|
| Method | PSNR | SSIM | ELPIPS | PSNR | SSIM | ELPIPS |
| Bi et al. | 22.46 | 0.933 | 0.720 | 23.62 | 0.951 | 0.518 |
| NeRV | 24.57 | 0.951 | 0.627 | 25.47 | 0.959 | 0.432 |
| Ours | **33.49** | **0.982** | **0.209** | **32.87** | **0.974** | **0.154** |

### F.2 Scenes trained on the "env + point"

Accordingly, Table 2 presents quantitative results for the *Bunny* and the *Buddha* scenes when they are trained with the "env + point" illumination. Each test view has a single novel point light. Our method numerically performs better than the baselines.

Table 2: Quantitative comparisons on the *Bunny* and the *Buddha* scene when they are trained with the "env + point" illumination. The PSNR ($\uparrow$), SSIM ($\uparrow$) and ELPIPS ($\downarrow$) [6] values are averaged over all images of a test set. ELPIPS values have a scale of $\times 10^{-2}$.

| Env+Point | Bunny | | | Buddha | | |
|---|---|---|---|---|---|---|
| Method | PSNR | SSIM | ELPIPS | PSNR | SSIM | ELPIPS |
| Bi et al. | 22.82 | 0.935 | 0.709 | 24.20 | 0.956 | 0.466 |
| NeRV | 25.24 | 0.959 | 0.597 | 27.36 | 0.968 | 0.345 |
| Ours | **32.93** | **0.980** | **0.293** | **32.74** | **0.976** | **0.204** |

## G Quantitative results for scenes of solid objects

In Table 3, we show the quantitative results on the *Dragon* scene and the *Armadillo* scene that contain glossy solid objects. Bi's method and the NeRV method use the parameter settings as described in Appendix D. Our method retains the same parameter settings as those for the participating media scenes. "Ours + BRDF" is a variant of our method that uses a BRDF function as the scattering function. We experimentally set its maximum spherical harmonic band to 1 and set the highest positional encoding frequency to $2^7$. Our method achieves higher quantitative metrics compared to the baselines and the variant.

Figure 8 of the main paper shows a comparison on one test view of the *Dragon* scene. Bi's method recovers the highlights on surfaces, but it produces an overexposed appearance and leads to faint shadows. Our method produces a smooth appearance and properly cast the shadow according to the novel lighting.

Table 3: Quantitative comparisons on two scenes with glossy solid objects. The scenes are trained with the "point" illumination and tested under novel lighting. The PSNR ($\uparrow$), SSIM ($\uparrow$) and ELPIPS ($\downarrow$) [6] are the averaged value over a test set. ELPIPS values have a scale of $\times 10^{-2}$.

| | Dragon | | | Armadillo | | |
|---|---|---|---|---|---|---|
| Method | PSNR | SSIM | ELPIPS | PSNR | SSIM | ELPIPS |
| Bi et al. | 19.60 | 0.897 | 1.358 | 19.19 | 0.897 | 1.243 |
| NeRV | 26.60 | 0.917 | 0.902 | 25.32 | 0.893 | 1.034 |
| Ours + BRDF | 28.32 | 0.931 | 0.692 | 26.46 | 0.917 | 0.847 |
| Ours | **28.50** | **0.942** | **0.564** | **26.60** | **0.924** | **0.753** |

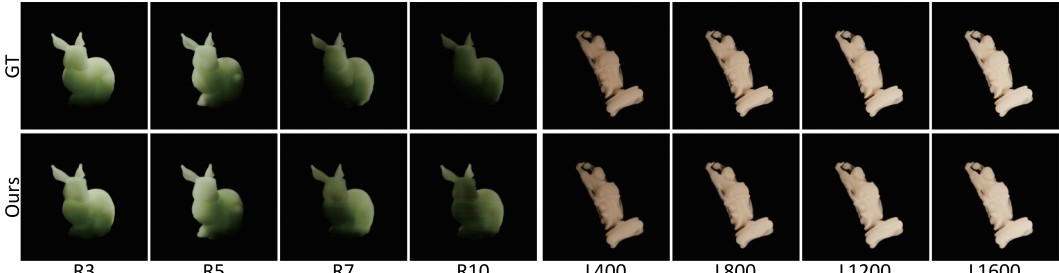

Figure 2: Qualitative comparisons of generalization on light location and light intensity.

| Bunny | Light distance (m) | | | | Buddha | Light intensity | | | |
|---|---|---|---|---|---|---|---|---|---|
| | 3 | 5 | 7 | 10 | | 400 | 800 | 1200 | 1600 |
| PSNR | 31.39 | 29.87 | 23.14 | 16.96 | PSNR | 32.82 | 33.04 | 31.91 | 28.47 |
| SSIM | 0.970 | 0.965 | 0.922 | 0.865 | SSIM | 0.982 | 0.984 | 0.981 | 0.974 |
| ELPIPS | 0.359 | 0.388 | 0.923 | 1.740 | ELPIPS | 0.204 | 0.195 | 0.219 | 0.235 |

Figure 3: Quantitative results for investigating the generalization on the light locations and light intensities. ELPIPS metrics have a scale of $10^{-2}$.

# H  Generalization quality

**Generalization quality for light distance.**  We used the *Bunny* scene in this experiment. During training, the point light's distance to the center of the bunny is stratified sampled from the range $(3, 5)$. We then build four test sets, each with 20 views; we set the point light's distance for the test sets as 3, 5, 7, and 10, individually. The light intensity is set to 600. We run the trained neural network on each test set. The image results are presented on the left of Fig. 2 and the numerical performance is shown in the left table of Fig. 3. The trained neural network performs relatively well when the test point light's distance is close to those in training, and the numerical performance gradually drops when the test point light moves away from the training manifold.

**Generalization quality for light intensity.**  We used the *Buddha* scene in this experiment. The training intensity values were stratified sampled from 50 to 900. We generate four test sets with the same 20 camera views. The point light is put at a distance 4 for all test sets. Also, we set the test light intensity as 400, 800, 1200, and 1600, respectively. Same as before, we run the trained neural network on each test set. The graphical comparisons and numerical metrics are shown on the right of Fig. 2 and Fig. 3. We observed that the trained neural network achieves high numerical performance when the test light intensity is within the range of the training intensity. For testing intensity outside of the training range, the numerical performance decreases.