# OpenReview forum: "Neural Relightable Participating Media Rendering"
_NeurIPS.cc/2021/Conference — NeurIPS 2021 Poster_

### Official Review · Reviewer_uER8 · 2021-07-12

**Rating:** 7
**Confidence:** 4

**Summary:**

This paper presents a neural volumetric inverse rendering approach for reconstructing a 3D representation of a scene with participating media (volumetric scattering) that can be rendered from novel viewpoints under novel lighting conditions. The paper builds on NeRV (an extension of NeRF that estimates relightable models of opaque objects) and modifies NeRV so that it can also recover relightable models of scenes with participating media. The main contributions of the paper are replacing the BRDF used in NeRV with a parametric phase/scattering function, and a technique to efficiently account for multiple scattering by training an MLP that predicts the "global illumination" at any location, represented by spherical harmonics, that is conditioned on a representation of the incoming lighting.

**Limitations And Societal Impact:**

The paper does a reasonable job discussing limitations, including limitations in the type of input data required and limitations in performance on certain types of scenes.

The paper does not discuss potential negative societal impacts. However, I do not think that this necessarily applies to the paper, because the paper addresses a more theoretical problem, and the recovery of relightable volumetric models from images does not seem to have any immediate safety/privacy/fairness concerns.

**Main Review:**

Strengths:

I think this paper is generally well-written, and the proposed method is logical and well-motivated. The experimental results validate the main idea of the paper that extending NeRV to handle participating media (by changing the BRDF to a scattering phase function and including a way to handle full indirect illumination) produces better results than existing techniques.

Weaknesses:

I'm a bit confused at the idea of conditioning the network that predicts spherical harmonics coefficients for multiple scattering on the binary e that represents the existence of environment lighting (lines 197-198). Shouldn't the multiple scattering depend on the actual environment lighting distribution, intensity, and color? How would the scene be rendered under arbitrary environment maps? Could the authors please discuss this?

I'm curious how this method would work when the asymmetry parameter g and volumetric albedo can also vary across the scene. For example, what if there are two volumetric objects with different properties next to each other?

Are there any results where the scattering albedo is not constant across the entire volume? I think that showing such experiments would make the visual results much more compelling and convincing. Page 7 mentions that the cloud scene is heterogeneous, but I think that choosing an example with more obvious albedo and asymmetry parameter changes across the scene would be more convincing.

I'm curious why "Ours+BRDF" seems to work worse than "Ours" in the dragon in Figure 6 where the material should be better modeled as a BRDF than as a phase function. Could the authors please discuss this more?

Minor edits/suggestions:

I would recommend changing the title to remove the word "Implicit", as this may be confusing since there is no implicit surfaces in this method. Maybe "Neural Volumetric Relightable Participating Media" could be better?

Line 41 mentions that this method learns a representation with volume density and scattering albedo. This might confuse readers by suggesting you are assuming a single known phase function, so I suggest clarifying this and maybe mentioning that the phase function parameter is optimized.

I would recommend not calling NeRF-like methods that use MLPs to represent continuous volumes "implicit". It makes more sense to call neural occupancy fields or SDFs "implicit" because they represent shapes as implicit functions (surfaces are level sets of a function), but NeRF is not trying to represent a single surface so I don't recommend calling it "implicit".

Line 68: most approaches --> most recent approaches

Line 79: the interest of the community --> of interest to the computer vision community

Line 83: based on the inverse rendering fashion --> based on differentiable pathtracing

Line 88: later enhancement --> extensions

Line 88: I'm not sure why just these two specific NeRF follow-ups are cited here, when there has been a great deal of extensions to NeRF (https://github.com/yenchenlin/awesome-NeRF plus even more).

Line 115: I'm not sure what "glosses" is supposed to mean here.

Line 190: fashion --> technique/strategy

I recommend proofreading the references section to fix references that are cited as "arXiv" but were actually published at other venues. For example, [9] was published at CVPR, [17] was published at SIGGRAPH, [23] was published at NeurIPS, etc.

[5], [9], and [41] should be capitalized as NeRF, NeRV, and NeRD in the references. [16] should be capitalized as DeepView. "3D" in [7] should be capitalized.

**Time Spent Reviewing:**

2

---

> ### Author Response · Authors · 2021-08-10
> **Response to Reviewer 4 (uER8)**
>
> ## Binary e for the envmap
>
> We used a single envmap for each scene when rendering the “env+point” datasets. For some camera views of a scene, the envmap lighting is on. For the rest of the camera views, the envmap lighting is off. In this setting, we found that a binary “e” is enough to indicate the existence of the envmap lighting. Also, the mentioned envmap distribution, intensity, and color could be used too, but they are not as concise as the binary “e”.
>
> ## How would the scene be rendered under arbitrary environment maps?
>
> Our method learns disentangled volume density and scattering properties of participating media from posed images. Afterward, we can leverage a volumetric path tracer to render the disentangled representation under arbitrary environment maps. Figure 5 demonstrates such renderings under arbitrary environment maps.
>
> ## Varying scattering albedo across the entire volume
>
> Thank you for suggesting this setting. We agree that albedo can also vary across the scene. Because our neural network predicts per-point albedo, our method can handle the participating media with varying albedo. We have built a scene with four different bunnies placed next to each other, each having a different albedo. Our method successfully handles the varying albedo and produces high-quality results. We will add the results in the final paper.
>
> ## varying asymmetry parameter g
>
> We have used the scene with four different bunnies placed next to each other, and configure each bunny with a different albedo and a different $g$. In this situation, we alter the neural network to also predict the per-point $g$. Our experiment results show that the neural network copes with the varying asymmetry parameter $g$ robustly. We will include the results in the final revision.
>
>
> ## "Ours+BRDF" seems to work worse than "Ours" in the dragon in Figure 6.
>
> We have carefully investigated this and found that the visibility network needs to see the rendering loss (left summand in Eq.9). This makes sense since the single scattering component computes the direct lighting which is affected by the presence of occlusion. This helps better learn the visibility and improve the lighting computation. We have tested this on the scenes of solid objects( Dragon and Armadillo) and report the error values below:
>
> |Dragon	|Bi et al.	| NeRV | Ours+BRDF	|Ours|
> |:-----------                 | :-----:      | :-----:       | :----------:          |:-----:     |
> |PSNR		|19.60	|26.60	|28.32		|28.50|
> |SSIM		|0.897	|0.917	|0.931		|0.942|
> |ELPIPS ($\times 10^{-2}$)	|1.358	|0.902	|0.692		|0.564|
>
> |Armadillo	|Bi et al.	| NeRV | Ours+BRDF	|Ours|
> |:-----------                 | :-----:      | :-----:       | :----------:          |:-----:     |
> |PSNR		|19.19	|25.32	|26.46		|26.60|
> |SSIM		|0.897	|0.893	|0.917		|0.924|
> |ELPIPS  ($\times 10^{-2}$)	|1.243	|1.034	|0.847		|0.753|
>
> These values are the mean values over all 30 test images. The final rendered results with “Ours+BRDF” perform better than the Bi et al. method and the NeRV method, both perceptually and numerically. “Ours” achieves slightly better numerical performance due to its smoother appearance. Yet, we observe from the images that “Ours+BRDF” perceptually preserves the glossy reflection on the dragon better than “Ours”, because the BRDF model is better in line with the material of the dragon.
>
> ## Suggestions of literal edits
>
> Thank you for the many valuable suggestions. We’ll take every effort to address all the edits as suggested.
>
> ## Change of title
> Thanks for the suggestion. We agree that the proposed title is more in line with the proposed approach.
>
> ## L115: “glosses”
>
> We meant that it ignores the underlying scattering properties and illumination. We will clarify this in the final version.
>
> ## Proofreading the references.
>
> We will fix the naming, capitalization, and publication venues of the references. We will also add other relevant NeRF-based papers.

---

> > ### Comment · Reviewer_uER8 · 2021-08-30
> > **Response**
> >
> > Thank you for the detailed response to my questions and concerns.
> > I appreciate the efforts to include results with varying albedo and asymmetry, and hope to see them in the updated draft.
> > I also share some of the concerns from Reviewer 8Z2s that the proposed method's using a network to predict the spherical harmonics approximation of indirect illumination/multiple scattering does not necessarily generalize well, and I would like to also see more experiments that probe the difference between rendering the model with the predicted spherical harmonics and using a volumetric pathtracer.
> > Overall, I still think that this is paper presents an interesting extension of previous "relightable NeRF" works to participating media, and I still think that it should be accepted.

---

> > > ### Author Response · Authors · 2021-09-01
> > > **Thank You Note**
> > >
> > > We appreciate your comments. We will incorporate the results with varying albedo and asymmetry parameter in the paper. Also, We will include the experiments (investigation on the SH prediction generalizability, comparison between rendering with the predicted SH and rendering with a volumetric path tracer) in the final version as supplementals.

---

### Official Review · Reviewer_2Q3Q · 2021-07-16

**Rating:** 7
**Confidence:** 3

**Summary:**

The authors present a neural implicit representation for participating media. This requires (i) extending neural radiance field approaches that are limited to a fixed lighting and therefore less flexible regarding relighting and scene editing, or (ii) overcoming methods for disentangling reflectance and illumination that are typically limited to solid objects without subsurface scattering and direct lighting. The presented approach takes input data in terms of a set of images with camera calibration data taken under varying but known lighting conditions. Then they use a NeRF-like approach for learning a disentangled representation for the participating media with volume density and scattering albedo. Here, they use a differentiable physically-based ray marching where they embed single scattering (using Monte Carlo ray tracing) and multiple scattering (using spherical harmonics approximation) separately to handle global illumination at reduced computational burden. This allows learning the decomposition of direct lighting and indirect lighting in an unsupervised manner.

In their evaluation, the authors demonstrate the learned representation of participating media to outperform related state-of-the-art techniques (in both quantitative and qualitative comparisons) and to be suited for scene relighting, scene editing and scene composition.


**Limitations And Societal Impact:**

The authors provide a discussion of limitations.

**Main Review:**

Originality
- The method seems novel and reasonable with plausible results for the datasets and offers benefits over state-of-the-art techniques regarding quality and inference time.


Evaluation
- The authors provide quantitative and qualitative comparisons for a diverse set of examples including homogeneous and inhomogeneous media.
- They also include related techniques that they adapted to handle scattering characteristics and show that their proposed approach outperforms these approaches.
- Further results address the editing of scene characteristics such as editing of albedo and density or scene compositing where the proposed approach provides reasonable results.
- Ablation studies address the impact of the number of spherical harmonics bands and exchanging the scattering function.
- The authors provide a discussion of limitations. Besides relying on known illumination and known camera poses, there are some further limitations. For solid glossy objects, the reconstruction preserves the base colors but faces difficulties in preserving the specular highlights. The consideration of refractive boundaries is left for future work. Still, this seems to be ok for the current submission.


Exposition
- The paper is well-structured and readable. Figures/tables and captions are informative.


Reproducibility
- The paper seems reproducible for a more experience person. The authors mention that they intend to release the code.


References
- References seem to be adequate.


Post-rebuttal comments:
I appreciate that the authors share further insights on other reviewers’ comments, report results for tests on scenes of solid objects and promise to release code and data. I remain at my previous recommendation "7: Good paper, accept".


**Time Spent Reviewing:**

4

---

> ### Author Response · Authors · 2021-08-09
> **Response to Reviewer 3 (2Q3Q)**
>
> Thank you for your comments.

---

> > ### Comment · Reviewer_2Q3Q · 2021-08-18
> > **Response**
> >
> > I appreciate that the authors share further insights on other reviewers’ comments, report results for tests on scenes of solid objects and promise to release code and data. I remain at my previous recommendation "7: Good paper, accept".

---

> > > ### Author Response · Authors · 2021-08-22
> > > **Thank You Note**
> > >
> > > We appreciate your encouraging comments.

---

### Official Review · Reviewer_8Z2s · 2021-07-17

**Rating:** 4
**Confidence:** 5

**Summary:**

The paper proposes a method to learn a volumetric representation for scenes with
participating media from a set of images. It builds on the NeRF architecture.
The difference part is that it applies a phase function to model how light is
scattered at point instead of using radiance field or BRDF. It also models the
incident multi-scatterings at each point using spherical harmonics to avoid path
tracing. They show better results than previous methods and demo applications   such as scene composition and editing.


**Limitations And Societal Impact:**

See the main review.

**Main Review:**


### Strengths
1. The usage of spherical harmonics to model incident lighting is technically sound,
and reduces the computation cost to account for multi-scatterings.

2. The results on synthetic data are better than NRF and NeRV, and better match the
ground truth.

### Weakness
1. The paper says that spherical harmonics can better represent the multi-scattering
terms. If that's the case, I would expect if the method changes the phase function to
BRDF, it would achieve better performance than NeRV, since the proposed method has
a more accurate lighting representation. However, from Figure 6, and the Figure at
Line 325, the proposed method is worse than NeRV. It make me wonder whether the proposed
multi-scattering representation is really beneficial. Why does the proposed method not
work well with BRDF even for solid objects?

2. At Line 280, the paper shows the images under single-scattering and multi-scattering.
Without giving the corresponding ground truth, it's difficult to interpret whether the
shown images are correct or not. In fact, I feel that the single-scattering is too dark.
For points at the surface, I would expect the single-scattering plays a dominant role.
However, it seems multi-scattering is much brighter than single-scattering. To really validate
this, ground truth images need to be provided.

3. While the proposed method estimates per-point albedo, all the synthetic scenes
have a constant albedo. How does the proposed method work for participating media with
varying albedo?

4. To predict the SH coefficients, the network takes in the point-light position and intensity
as input. This does not make sense to me. I don't think such a design can generalize to
arbitrary lighting intensity and position, considering that you only have a single set of
lighting condition for the training images?
Also what if you have multiple point lights with an envmap in training?  In fact, it's not clear to
me how the proposed method performs the testing.  How is the multi-scattering rendered
during relighting? Given a new lighting condition, the SH for multi-scattering
would change, and the originally predicted SH cannot be used. Is the paper using path tracers
to generate images in Figure 3, 4? Or the SH are re-predicted with the new lighting condition?

5. For the renderings with Mitsuba, the objects such as the bunny and the buddha are more
translucent than they should be. What is the reason?

6. The qualitative results are not fully convincing. The generated results tend to be blurry and
missing details, also the shading looks incorrect when the light is at an grazing
angle (video 02:28). While the paper acknowledges the ringing artifacts, such artifacts
also limit the contributions of the paper.

Overall, while I agree the usage of SH for multi-scatterings is reasonable, I am doubtful about
the design of the network to predict the SH and whether it can generalize to arbitrary lighting
position and intensity. The quality of the results also needs further improvement, and more
validations against GT under more diverse lighting conditions are also needed.



**Time Spent Reviewing:**

2

---

> ### Author Response · Authors · 2021-08-10
> **Response to Reviewer 2 (8Z2s)**
>
> ## Why does the proposed method not work well with BRDF for solid objects?
>
> We have carefully investigated this issue and found that the visibility network needs to see the rendering loss (left summand in Eq.9). This makes sense since the single scattering component computes the direct lighting which is affected by the presence of occlusion. This helps better learn the visibility and improve the lighting computation. The final results with “Ours+BRDF” perform better than the Bi et al. method and the NeRV method, both perceptually and numerically. We have tested this on the scenes of solid objects (Dragon and Armadillo) and report the error values below:
>
> |Dragon	|Bi et al.	| NeRV | Ours+BRDF	|Ours|
> |:----------                 | :-----:      | :-----:       | :---------:          |:-----:     |
> |PSNR		|19.60	|26.60	|28.32		|28.50|
> |SSIM		|0.897	|0.917	|0.931		|0.942|
> |ELPIPS ($\times 10^{-2}$)	|1.358	|0.902	|0.692		|0.564|
>
> |Armadillo	|Bi et al.	| NeRV | Ours+BRDF	|Ours|
> |:----------                 | :-----:      | :-----:       | :---------:          |:-----:     |
> |PSNR		|19.19	|25.32	|26.46		|26.60|
> |SSIM		|0.897	|0.893	|0.917		|0.924|
> |ELPIPS ($\times 10^{-2}$)	|1.243	|1.034	|0.847		|0.753|
>
> These metric values are the mean values over all 30 test images.
>
> For the bunny scene (L325), the above change in the visibility network further improves the rendering quality, however, some artifacts can still be seen. We also compared it against the NeRV pipeline (BRDF), but NeRV’s result is even worse. We will be happy to add these new results to the final version.
>
> ## Ground truth for the L280 inline figure: single-scattering is too dark
>
> From the ground truth images, we found that the multiple-scattering component plays a dominant role in the appearance of the participating media. To confirm this, we rendered the ground-truth single (direct) and multiple (indirect) scattering components individually: both the predicted single-scattering and the predicted multiple-scattering have a comparable brightness to their ground truth images, respectively. We will provide the ground-truth single-scattering images and multi-scattering images in the supplemental document in the final version.
>
> ## How does the proposed method work for participating media with varying albedo?
>
> Thanks for suggesting this setting. Because our neural network predicts per-point albedo, our method can handle the participating media with varying albedo. We have built a scene with four different bunnies placed next to each other, each having a different albedo. Our method successfully handles the varying albedo and produces high-quality results. We will add the results in the final paper.
>
> ## What if multiple point lights with an envmap in training?
>
> Our current setups do not take into account multiple point lights with an envmap. “Multiple fixed point lights + a random point light” was investigated in NeRV (Srinivasan et al.) by using a PointNet encoder to embed the position and intensity of every light source. Our processing of the lighting condition can be augmented with this strategy and the envmap can be processed by learning an environment embedding. This would be an interesting point for future extension.
>
> ## Does it generalize to arbitrary lighting intensity and position?
>
> Our training was performed on two different lighting conditions for each scene:
> * “point”: For each camera view, we have a random point light source.
> * “env+point”: For each camera view, we have a random point light source + a fixed environment lighting. The existence of the environment lighting is marked by a binary indicator.
>
> After training, our method can generalize for novel point light locations and novel intensities. It renders novel views with the neural network using ray marching. While the rendering does not generalize for exotic lighting conditions, e.g., different shaped-light sources, our learned disentangled representation with volume density and albedo still allows a way to render images under arbitrary lighting conditions. As demonstrated in Figure 5, we can render the learned volume density and albedo under novel environment lighting.
>
>
> ## It's not clear to me how the proposed method performs the testing.
>
> During testing, the single scattering (direct lighting) component is computed by tracing the rays from the camera ray samples to the test light sources. For multiple scattering, the light position, intensity, and envmap indicator are sent to the SH network for predicting the SH coefficients. The computed single scattering component and multiple scattering component are combined to form the final image.
>
> ### The SH coefficients are re-predicted with the new lighting condition?
>
> Yes.
>
> ### How is the multi-scattering rendered during relighting?
>
> We use the re-predicted SH to compute the multi-scattering part.
>
> ## Is the paper using path tracers to generate images in Figure 3, 4?
>
> No, they are rendered by the trained neural network using ray marching. Note that, path tracing requires explicit and refined geometry information which is not accessible in our case.
>
> ## Bunny and the buddha are more translucent than they should be. What is the reason?
>
> We have re-rendered the scene on the right of Figure 5 (L303) with the environment lighting of the left image. From the rendered image, we verify that the bunny has the same translucency as in the left image.
>
> ## The qualitative results are not fully convincing
>
> We agree that the quality may not be comparable to physically-accurate participating media rendering, which has explicit geometry and scattering properties. Note that our method has no access to the ground truth geometry and scattering properties. Instead, it learns the volume density and scattering properties of participating media merely from posed images.
> Further, there is no existing benchmark for neural participating media, which makes it harder to qualify the results. Compared to the current state-of-the-art methods, our approach demonstrates fewer artifacts and faster inference speed when performing participating media rendering. To help future work to improve upon this, we will release our dataset and the code.

---

> > ### Comment · Reviewer_8Z2s · 2021-08-21
> > **Reviewer feedback**
> >
> > Thank the authors for the rebuttal. I still have questions about predicting SH under novel lighting conditions during testing.
> >
> > As the authors say, the SH is re-predicted with the network that takes the light position, intensity, and envmap as input. As the paper says at line 258, "the first one “point” has a white point 258 light with varying intensities sampled within 50 ∼ 1200 and its location is randomly sampled on 259 a sphere over the scene. So it means that the network can only predict the SH when the light is on the sphere and its intensity is in the training range. However, in practice, the lighting position can be arbitrary in the space, and the intensity can also be arbitrary. It's impossible for the network to be scalable to an arbitrary single point relighting. I hope the authors can further clarify this.

---

> > > ### Author Response · Authors · 2021-08-22
> > > **Response to the feedback**
> > >
> > > We are glad to clarify this. After training, we can render the learned participating media in two ways.
> > > 1. Render with the neural network using ray marching (Figure 3 & 4):
> > > In this situation, we agree that both the intensities and locations of the point light can vary continuously. Also, they may have an infinite number of values. Therefore, we did not claim that the rendering with the network and SH can scale to a point light with an arbitrary intensity and location.
> > >
> > >
> > >     The specific light intensities and light locations for our training set and the test set are as follows,
> > >     #### __Training set__
> > >     * Light intensity: ranges from 50~1200
> > >     * Light locations: located on a sphere with a radius 4
> > >
> > >     #### __Test set__
> > >     * Light intensity: ranges from 50~1500, and do not coincide with intensity values of the training set
> > >     * Light locations: located on a sphere with a radius 3 and a sphere with a radius 5
> > >
> > >
> > >     In the above settings, the test set ensures the light intensity and light locations are different from those of the training set. SH is re-predicted for each test view. Our evaluations on the test data demonstrate that the trained neural network can generalize to the light intensities and locations of the test set. Thereby, the generalization is in an interpolation manner. We will add discussion of the generalizability of our approach in the revision and include the details of the training set and test set.
> > >
> > > 2. Rendering with volumetric ray tracing by a rendering engine (Figure 5):
> > > Our approach is able to learn a disentangled representation of participating media from posed images with known lighting conditions. Our learned disentangled representations allow the predictions to be used in rendering with volumetric ray tracing by a rendering engine. This is done by querying the trained neural network to obtain volume density and albedo. Then, we can render it under a novel lighting condition, such as an arbitrary point light, using a rendering engine. Please note also that SH is no longer computed here, and the illumination computation will be processed by the rendering engine.

---

> > > > ### Comment · Reviewer_8Z2s · 2021-08-27
> > > > **Reviewer feedback**
> > > >
> > > > I thank the authors for the clarification. I have the following questions:
> > > >
> > > > 1. Why can the SH prediction network generalize to point light locations outside of the training manifold? If the network only sees point lights on a sphere of radius 4, how can it predict the SH under the radius of 3 and 5? Similar question for intensity.
> > > >
> > > > 2. While I understand both methods can be used to render, the authors don't show whether these two would match or not. If the scene is rendered with both methods under the same lighting, would the results be the same? In addition, the paper shows rendering with path tracers without providing ground truth, so it's difficult to judge whether the renderings with path tracers are correct or not.

---

> > > > > ### Author Response · Authors · 2021-09-01
> > > > > **Response**
> > > > >
> > > > > ## SH prediction generalizability for light locations and light intensity
> > > > > To investigate this issue, we have conducted two experiments:
> > > > >
> > > > > ### 1. Analyzing the generalization quality for light locations (radius in our setting)
> > > > >
> > > > > In this experiment, we used the bunny scene. The neural network was trained with the point lights residing on a sphere of a radius 4. To build the test sets, we took 20 test views, put new point light on spheres of radius 3, 5, 7, 10, 30, 50, and set the light intensity to 600. For each radius, we render a set of 20 images. Then, we run the trained neural network on each test set. And, we have obtained the numerical performance for these radius choices,
> > > > >
> > > > > |Radius|3|5|7|10|30|50|
> > > > > |:------|:------:|:------:|:------:|:------:|:------:|:------:|
> > > > > |PSNR$\uparrow$|31.39|29.87|23.14|16.96|14.78|13.26|
> > > > > |SSIM$\uparrow$|0.970|0.965|0.922|0.865|0.784|0.773|
> > > > > |ELPIPS($\times10^{-2}$$\downarrow$)|0.359|0.388|0.923|1.740|3.794|4.095|
> > > > >
> > > > > These metric values are the mean values over the 20 test images.
> > > > >
> > > > > From this experiment, we observed that the trained neural network performs relatively well when the test radiuses are close to the training radius, and the numerical performance drops when the test light moves far away. In summary, our observation is that the generalization quality of the neural network gradually decreases when the test light moves away from the training manifold.
> > > > >
> > > > > ### 2. Analyzing the generalization quality for light intensity
> > > > >
> > > > > In this experiment, we used the buddha scene and generated 20 test views. We put the test point light on a sphere of radius 4 and kept the 20 camera poses unchanged. This time we set the test light intensity as 400, 800, 1200, 1400, 1600, 2000. By contrast, the training intensity values were sampled from 50 to 1200, but did not coincide with 400, 800, and 1200.
> > > > > For each testing light intensity, we render a set of 20 images. Same as before, we run our trained neural network on each test set. And, we have obtained their numerical performance,
> > > > >
> > > > > |Intensity	|400	| 800	 |1200	|1400	|1600	|2000|
> > > > > |:------|:------:|:------:|:------:|:------:|:------:|:------:|
> > > > > |PSNR$\uparrow$|32.82|33.04|31.91|28.14|28.47|27.36|
> > > > > |SSIM$\uparrow$|0.982|0.984|0.981|0.973|0.974|0.970|
> > > > > |ELPIPS($\times10^{-2}$$\downarrow$)|0.204|0.195|0.219|0.244|0.235|0.280|
> > > > >
> > > > > These metric values are the mean values over 20 test images.
> > > > >
> > > > > We observed that the trained neural network achieves good performance when the test intensity is within the range of the training intensity. For testing intensity values outside of the training range, the numerical performance decreases.
> > > > >
> > > > > ## Ray marching neural network vs. volumetric path tracer under same lighting conditions
> > > > >
> > > > > We have conducted an experiment to compare rendering with ray marching and rendering with the volumetric path tracer from mitsuba. We used the bunny scene in this experiment. The lighting and camera poses are the same for the two methods. To render with the volumetric path tracer, we first queried the trained neural network to obtain two 128 x 128 x 128 data volumes, one for volume density and the other one for albedo. The maximum path depth for the volumetric path tracer is set to 30. We get the numerical performance of the rendered test view as
> > > > >
> > > > > |Method|Neural network | Path tracer|
> > > > > |:--------| :-------:|:-------:|
> > > > > |PSNR$\uparrow$|30.88|29.91|
> > > > > |SSIM$\uparrow$|0.971|0.968|
> > > > > |ELPIPS($\times10^{-2}$$\downarrow$)|0.269| 0.327|
> > > > >
> > > > > These values are the mean values over 20 views.
> > > > >
> > > > > Visually the images rendered with a volumetric path tracer (using queried volume data) look similar to the ones directly rendered using our neural network (using ray marching). Both results are also visually similar to the ground truth. We will be happy to add the corresponding rendered images in the final version.
> > > > >
> > > > > ## Ground truth images for path-traced renderings
> > > > >
> > > > > We have rendered the ground truth images for the scenes shown in Figure 5. The maximum path depth for the volumetric path tracing is set to 30. We evaluated the results that use our neural representation and render with a volumetric path tracer, against the ground truth. The PSNR/SSIM/ELPIPS metrics for the left scene are  28.46/0.942/0.0064, and the PSNR/SSIM/ELPIPS metrics for the right scene are 27.73/0.936/0.0072. We will add the ground truth images in the supplemental.

---

> > > > > > ### Comment · Reviewer_8Z2s · 2021-09-02
> > > > > > **Reviewer feedback**
> > > > > >
> > > > > > I thank the authors for all the efforts and rebuttals. However, after reading all the responses, I have to say I don't think the paper is suitable for acceptance in the current form. The SH prediction network is the major contribution of the paper, but its design inherently suffers from poor generalization. While path tracer can be used to render, it is only used as an application without thorough evaluations in the paper. In addition, I also share other reviewers' concerns on the lack of results on real data. All these factors make me refrain from improving my scores.

---

> > > > > > > ### Author Response · Authors · 2021-09-02
> > > > > > > **Response:**
> > > > > > >
> > > > > > > We would like to emphasize that our method makes several significant contributions.
> > > > > > > 1. Firstly, our novel SH design can efficiently cope with multiple-scattering, which is not possible for compared methods (NeRV and Bi et al.) . For example, Bi et al. simulates simply the direct illumination, NeRV copes with direct illumination and one-bounce indirect illumination. Our method handles the complete illumination with both direct illumination and indirect illumination in a principled way.
> > > > > > >
> > > > > > > 1. Second, our method shows the demonstration to learn a disentangled neural representation for participating media.
> > > > > > >
> > > > > > > 1. Third, our method demonstrates higher visual quality and faster inference speed compared to state-of-the-art methods.
> > > > > > >
> > > > > > > 1. Fourth, our work has established a synthetic training and evaluation dataset with photorealistic images of the participating media, and our method overtakes the comparisons on this dataset. Additionally, we demonstrate promising applications including relighting, scene editing and scene composition. Further, no prior work has shown fast, high-quality neural rendering of participating media and there is also no existing dataset for the problem.
> > > > > > >
> > > > > > > To summarize, our method demonstrates a successful proof-of-concept for learning neural representation for participating media. We believe it represents a crucial step towards learning from real data.
> > > > > > >
> > > > > > > We will provide all the rendering results to support our evaluations. Our rendered results are visually similar to the ground truth: both using the ray marching approach and with a volumetric path tracer on the queried data. The code and the dataset will be made public.

---

### Official Review · Reviewer_wnsz · 2021-07-30

**Rating:** 7
**Confidence:** 4

**Summary:**

This paper proposes a (inverse) neural rendering method for participating media that can disentangle volume density, scattering albedo and illumination from posed images and known light positions. Compared with previous NeRF methods, such as NeRF and NeRV, this approach can better deal with scattering media, such as fog and wax, in the meanwhile, generalizing to solid material.

**Limitations And Societal Impact:**

See weaknesses in main review.

**Main Review:**

### Strengths:

1. As for (inverse) neural rendering for participating media, the learning-based problem formulation is novel, and this method can jointly estimate volume density, scattering albedo and lighting in an unsupervised manner.

2. Compared with previous NeRF-based methods, the proposed method can capture more accurate global illuminations of participating media.

3. A new synthetic training and evaluation dataset for participating media is proposed.


### Weaknesses:

1. My major concern about this paper is lack of experiments on real data, and the authors mentioned this in limitations as well. Because the real-world environment and material are much more complex than simulation, it is hard to tell how well this method can generalize to real-world scenes.

2. The writing needs polish, e.g., the authors listed four contributions in L53-60, but they are basically saying the same thing, and I think the 2nd and 3rd ones can be merged. As for the 4th contribution, the authors can emphasize that the proposed method outperforms previous arts. In addition, L203 mentioned parameter g in Eq. 3, but I did not find g in Eq.3.

**Time Spent Reviewing:**

3

---

> ### Author Response · Authors · 2021-08-10
> **Response to Reviewer 1 (wnsz)**
>
> ##  Experiments on real data
>
> As the discussion in the limitations section (Section 6), this is definitely an interesting point. We are glad to share more thoughts on it. Acquiring real data of participating media involves many other concerns. For natural phenomena like cloud and fog, suitable acquisition hardware will be used to capture these phenomena at a feasible location and time. For small-scale participating media objects, we could pick objects, like milk, wax, juices, which have subsurface scattering. In this case, the lighting and the camera poses will need to be estimated. Extending our method to handle these cases would be an interesting future research direction.
>
> ## Polish (merge and modify) the contributions
>
> We agree with the suggestions and will incorporate them in the final version of the paper.
>
> ## L203 parameter g of Eq. 3
>
> The $g$ is a parameter of the HG phase function $\rho$ in Eq. 3 and the equation of the HG phase function is in Section A of the supplemental document. To make it explicit, we will add the HG phase function in the main paper.
>
> ## A new synthetic training and evaluation dataset for participating media is proposed
>
> We will release the dataset and the code.

---

### Author Response · Authors · 2021-08-10
**Response general**

We thank reviewers for their feedback. We are encouraged by comments like “method is novel, logical, well-motivated (R1, R3, R4)”, and “the usage of SH is technically sound and reasonable (R2)”. As pointed out by the reviewers, our method is able to learn a disentangled neural representation for participating media with volume density and scattering albedo, it can better deal with multi-scattering using SH at a reduced computational cost, it allows learning the decomposition of direct lighting and indirect lighting in an unsupervised manner, and it offers benefits for neural participating media rendering and outperforms state-of-the-art methods regarding the quality and inference time. Additionally, we demonstrated the applications including relighting, scene editing, and scene composition. Therefore, we believe this approach will provide inspiration and spur for follow-on work. Our responses to every review are posted below each review section.

---

### Decision · Program_Chairs · 2021-09-27

**Decision:**

Accept (Poster)

**Comment:**

The reviews were split: 3 reviewers gave 7 and one gave 4. Those who supported acceptance acknowledge the novelty of the paper as the method can jointly estimate volume density, scattering albedo and lighting in an unsupervised manner, and can capture more accurate global illuminations of participating media. The reviewer who gave 4 had concerns that he proposed method's using a network to predict the spherical harmonics approximation of indirect illumination/multiple scattering does not necessarily generalize well, and there was lengthy discussion between the authors and this reviewer, who was not convinced eventually. The AC agrees with the majority that the paper presented an interesting extension of previous "relightable NeRF" to participating media. The novelty and results warrant acceptance.